# Diagnosing Hemophagocytic Lymphohistiocytosis with Machine Learning: A Proof of Concept

**DOI:** 10.3390/jcm11206219

**Published:** 2022-10-21

**Authors:** Thomas El Jammal, Arthur Guerber, Martin Prodel, Maxime Fauter, Pascal Sève, Yvan Jamilloux

**Affiliations:** 1Internal Medicine, University Hospital Croix-Rousse, Hospices Civils de Lyon, 69004 Lyon, France; 2Independent Researcher, 69006 Lyon, France; 3Research on Healthcare Performance (RESHAPE), INSERM U1290, Université Claude Bernard Lyon 1, 69004 Lyon, France; 4Lyon Immunopathology Federation (LIFE), Hospices Civils de Lyon, 69000 Lyon, France

**Keywords:** hemophagocytic lymphohistiocytosis, inflammation, machine learning, Hscore, HLH-2004

## Abstract

Hemophagocytic lymphohistiocytosis is a hyperinflammatory syndrome characterized by uncontrolled activation of immune cells and mediators. Two diagnostic tools are widely used in clinical practice: the HLH-2004 criteria and the Hscore. Despite their good diagnostic performance, these scores were constructed after a selection of variables based on expert consensus. We propose here a machine learning approach to build a classification model for HLH in a cohort of patients selected by glycosylated ferritin dosage in our tertiary center in Lyon, France. On a dataset of 207 adult patients with 26 variables, our model showed good overall diagnostic performances with a sensitivity of 71.4% and high specificity, and positive and negative predictive values which were 100%, 100%, and 96.9%, respectively. Although generalization is difficult on a selected population, this is the first study to date to provide a machine-learning model for HLH detection. Further studies will be required to improve the machine learning model performances with a large number of HLH cases and with appropriate controls.

## 1. Introduction

Hemophagocytic lymphohistiocytosis (HLH) is a hyperinflammatory syndrome characterized by uncontrolled activation of T cells and macrophages, triggered by a genetic (primary) or acquired (secondary) cause [1,2]. When HLH occurs in the setting of an immune-mediated inflammatory disease (IMID) it is consensually termed macrophage activation syndrome (MAS) [3]. The pathogenesis of HLH may involve a defect in lymphocyte cytotoxicity or an overactivation of inflammatory pathways resulting in a genuine cytokine storm. In secondary HLH (sHLH), the trigger can be of several natures: infections (e.g., Epstein Barr virus, EBV; cytomegalovirus, CMV; tuberculosis, etc.); hematological malignancies, IMIDs (mainly Still’s disease and systemic lupus erythematosus); and other conditions including drug reactions and heavy surgeries (e.g., post-transplant) [4,5].

HLH diagnosis in adults relies mainly on the clinical expertise of physicians, although the classification criteria proposed by The Histiocyte Society (HLH-2004) are regularly used as a reference in studies [6]. However, these criteria were developed for children with primary HLH and have not been formally validated for sHLH or adults [7]. In addition, these criteria may hamper early diagnosis as some criteria (e.g., cytopenias and hemophagocytosis) may occur late and some tests (e.g., NK cell activity and sCD25 levels) are not routinely available in hospital laboratories [1]. Finally, although considered a hallmark of HLH, hemophagocytosis on bone marrow smears is not specific and can be found in up to 2/3 of patients without HLH [8,9].

The median delay to HLH diagnosis is 10 days [10,11]. The overall mortality rate of sHLH is 40% with a 30-day mortality of 20% [10]. Therefore, prompt recognition of HLH is mandatory. Several diagnostic or classification scores have been proposed. The PRINTO criteria (MAS score-05) and then the classification criteria (MAS score-16) as well as the MAS/sJIA score (MS score) are to be used for children with MAS complicating systemic juvenile idiopathic arthritis (sJIA) [12,13,14]. The Hscore was developed in adults with sHLH but the original study included a very limited number of patients with IMIDs (3%), while the vast majority of patients had hematological malignancies [15]. Several subsequent studies aimed at testing the validity of the Hscore on cohorts of patients with more varied triggers have all found poorer diagnostic performance. In particular, specificity was variable (specificity between 71–93%, depending on the study vs. 86% in the original population) [7,15,16,17]. In total, none of these scores are completely satisfactory for the diagnosis of sHLH and none have achieved complete consensus. Moreover, the external validation in these studies is limited by their retrospective nature. Rather than the static view provided by the classification criteria, some authors have proposed a dynamic study of different clinical and biological parameters in HLH [18]. In this perspective, several biomarkers (e.g., ferritin, glycosylated ferritin, IL-18, sIL-2R, etc.) have been evaluated but none of them have reached sufficient accuracy to be of diagnostic value in HLH [6,19]. It now seems clear that a reductionist approach to a single parameter will not achieve an interesting specificity.

Over the last decade, artificial intelligence has become a powerful tool to assist and improve medical decision making, mainly through deep learning models in various specialties (e.g., radiology, dermatology, ophthalmology) [20]. Yet, no machine learning model is currently available for HLH diagnosis. Our study aimed to build a reliable machine-learning model for the diagnosis of HLH in adults.

## 2. Materials and Methods

### 2.1. Study Design and Population

We used data from a French cohort, which we previously analyzed, to investigate the diagnostic performance of glycosylated ferritin (GF) in HLH and adult-onset Still’s disease (AOSD) [21]. In brief, we conducted a retrospective, case-control study by collecting data from all adult patients who had at least one GF measurement between 1 January 2018 and 31 December 2019 (before the COVID-19 pandemic). The institutional biochemistry laboratory in Lyon, France, where the assays were performed, receives samples from five French university hospitals. The institutional review board approved the study protocol (#22-862). In accordance with French legislation on non-interventional retrospective studies, no written informed consent was required for inclusion.

### 2.2. Data Classification

All these methods were previously reported [21]. Biological data were automatically extracted from two electronic patient data management systems (GLIMS9-MIPS (Clinisys, Gent, Belgium), and EASILY-HCL (Hospices Civils de Lyon, Lyon, France)). Clinical data were collected using a standardized form. The study period was defined as the time between the GF measurement and the last available data in the medical record.

The diagnosis of HLH was based on the final opinion of the clinician (i.e., compatible clinicobiological presentation, disease course, treatments, and management). We systematically collected on our forms the presence of a hemophagocytosis pattern if a pathology test was performed (within bone marrow, spleen, or lymph node) and the Hscore [15] was calculated retrospectively to assist in the classification of patients but was not mandatory to retain the diagnosis. The diagnosis was retained if: (i) it was confirmed by the referring physician at the last follow-up; (ii) it was validated by two independent investigators (YJ and AG), with conflicting cases discussed with a third expert (PS) and finally classified by consensus. Differential diagnoses were always sought and ruled out. Controls included patients who did not meet the predefined definition of HLH.

Six etiological subgroups were considered, whether in the HLH or in the control group: infectious diseases, solid cancers, hematological malignancies, immune-mediated inflammatory diseases (IMIDs), acute hepatitis, and a group encompassing all other less frequent conditions. The classification of patients into each subgroup was based on the diagnosis made by the treating physician. Each case was reviewed by the two independent investigators (YJ and AG) and confirmed by a third expert (PS), if necessary. In the case of residual uncertainty, cases were excluded.

### 2.3. Predictive Model Building

Data preprocessing was performed with R version 4.1.2. (R core team, Vienna, Austria) and model building was performed with Python version 3.9.7. (Centrum voor Wiskunde en Informatica, Amsterdam, The Netherlands) using the scikit-learn version 1.0.2 (Inria, Paris, France) [22]. The machine learning procedure was performed with an 11th Gen Intel^®^ Core™ i7-11850H at 2.50 GHz with 32 GB of RAM. No graphics processing unit was used in this work.

The dataset was first described by comparing the population with and without HLH. Categorical variables were described as mean and percentages when following a normal distribution and with medians and interquartile ranges when not. Categorical variables compared using a Fisher’s test (if one of the contingency table’s cells was equal or below 5) or using a Chi^2^ test. Continuous variables were described as means and standard deviations when normally distributed and compared with a Student’s *t*-test or as medians and interquartile ranges and compared with a Wilcoxon signed rank test when they followed a skewed distribution.

In the first step, we kept only the variables with less than 20% missing values. The remaining missing values were imputed using multiple imputations with a chained equations algorithm using the classification and regression trees method from the R package mice [23]. We then subtracted from the remaining variables all treatment-related variables or variables that could be collected after the diagnosis of HLH (e.g., death and treatments), thus not relevant for predicting HLH.

The final dataset consisted of 26 variables for each patient, with 10 being categorical (sex, hepatomegaly, splenomegaly, lymph node enlargement, immunosuppression, known genetic predisposition to HLH, human immunodeficiency virus (HIV) infection, hematological malignancy, solid neoplasia, and liver disease) and 16 continuous (age, body mass index (BMI), ferritin, glycosylated ferritin, C-reactive protein, aspartate aminotransferase (AST), alanine aminotransferase (ALT), total bilirubin, creatinine, prothrombin rate, hemoglobin concentration, platelet count, leukocyte count, neutrophil count, lymphocyte count, and maximal body temperature) variables. Due to performance loss, no scaling procedure was applied for continuous variables and categorical variables were encoded 0 if the criterion was absent (or if male sex) and 1 if present. Again, due to performance loss, no variables among the 26 were removed from the final model.

The dataset was then randomly split into training and test sets with respectively 70% and 30% of the initial population. The resulting training dataset was strongly imbalanced with 10% of patients with HLH. In addition, as the total number of observations was rather small from a machine learning perspective, a synthetic minority oversampling technique (SMOTE) was used on the training set to improve the upcoming model performances [24].

Regarding model selection, there is no consensus nor simple way to anticipate which machine learning algorithm will perform best on a specific prediction task [25]. For that reason, we have implemented a three-step benchmark of the most widespread algorithms.

Step 1: In order to quickly establish the relevance of using a machine learning approach and to decipher the most performant type of algorithm, we used an automated machine learning model from autoML to run naive (i.e., non-tuned) algorithms. This step was performed through the MLJAR-autoML tool [26]. We chose to run nine different algorithms to investigate a large variety of patterns: a random forest, an extreme gradient boosting (XGBoost), a decision tree classifier, a light gradient boosting, an extra-trees, a CatBoost, a linear model-a K-nearest neighbors, a neural network (although neural networks are not expected to perform well on datasets with only hundreds of observations), and a final ensemble model which consists in a linear ponderation of all the previous classifiers. A baseline algorithm, always predicting the majority class (i.e., “without HLH”), was used as a reference and performed in an 0.5 area under the curve (AUC) by definition. At this stage, the AUC was used to evaluate the algorithms’ performances, rather than accuracy, since data were imbalanced (more patients without HLH than with HLH) and we were more interested in predicting well HLH patients.

Step 2: The three most performing algorithms according to AUC were chosen to be further configured. For the configuration, tuning of their hyperparameters followed a grid search with a five-fold cross-validation procedure (80–20% ratio) for which optimization was based on the AUC metric.

Step 3: Finally, the three algorithms were trained altogether with a stacking classifier [25], with the meta-estimator being an XGBoost classifier. Stacking has been proven to lead to increased performance [27]. To compare the different models on the test set, sensitivity, specificity, positive predictive value, negative predictive value, accuracy (true positive (TP) + true negative (TN)/(positive (P) + negative (N)), and F1-scores (2 × TP/(2 × TP + false positive + false negative)) were calculated. In order to ease the understanding of how the final model dealt with variable weights, Shapley values were computed with a Kernel Explainer from the Shap library [28]. Shapley values represent the contribution of a feature to the model in a specific condition (i.e., a specific patient). For linear models, Shapley values are equal to the weight of the feature in the model times the value of the feature. In non-linear models, Shapley values represent the contribution of a feature depending on all other features in the model (e.g., in HLH detection for a 68 year old male patient with splenomegaly and immunosuppression, the Shapley value of splenomegaly for this patient represents the contribution of the feature “splenomegaly” to predict HLH given an age of 68, male sex, and immunosuppression).

## 3. Results

### 3.1. Descriptive Statistics

Considering age at disease onset and sex, there were no significant differences between the HLH and the control group (Table 1). Nearly half the patients in the HLH group had hematological malignancy at the time of data collection while patients in the control group were more frequently experiencing inflammatory disorders (*p* < 0.001). Maximal body temperature was higher in the HLH group (*p* < 0.001) as well as ferritin levels (*p* < 0.001). Patients with HLH had also more frequent enlarged lymph nodes, splenomegaly, and hepatomegaly (*p* < 0.05). Glycosylated ferritin was significantly lower in the HLH group (23.5 vs. 39.0, *p* < 0.001). The description of other variables from the original dataset is provided in Table 1.

### 3.2. Running the Automated Machine Learning Procedure

First, we ran a naive automated machine learning model (autoML) without hyperparameter tuning in order to decipher which algorithm types would best fit the dataset. Since the initial problem is a binary classification problem, we ran the autoML algorithm with classifiers and metrics on the train set are displayed in Table 2.

The ensemble model had an AUC metric of 0.966 on the test set (Figure 1A). The model yields a sensitivity (recall) and specificity of respectively 71.4% and 98.4% on the test set (Table 3 and Figure 2). The positive predictive value/precision (PPV) and the negative predictive value (NPV) were respectively 83.3% and 96.9%. The F1-score and accuracy were respectively 76.9% and 95.7%. The total autoML procedure took 19.4 s.

### 3.3. Hyperparameter Tuning and Fitting of the Ridge Classifier

Among the top three machine learning (ML) algorithms with the best AUC in the autoML procedure, we first fit a linear model with a ridge classifier. The ridge classifier after hyperparameter tuning had an AUC of 0.930 on the test set (Figure 1B). The model had a sensitivity, specificity, PPV, and NPV of 100%, 87.3%, 46.7%, and 100%, respectively (Table 3 and Figure 2). Furthermore, the F1-score and the accuracy were respectively 63.6% and 88.6%. Hyperparameter tuning took 24.7 s and the best model fit took 14 ms.

### 3.4. Hyperparameter Tuning and Fitting of the CatBoost Classifier

The second-ranked machine learning algorithm in terms of AUC in the autoML procedure was the CatBoost classifier. Like the ridge classifier, a five-fold cross validation was performed on a grid search for hyperparameter tuning. The AUC was 0.934 (Figure 1C). The sensitivity, specificity, PPV, and NPV were 71.4%, 95.2%, 62.5%, and 96.8%, respectively (Table 3 and Figure 2). The F1-score and the accuracy were respectively 66.7% and 92.9%. The grid search cross validation took 2 min and 33 s and it took 453 ms for the best model to fit.

### 3.5. Hyperparameter Tuning and Fitting of the K Nearest Neighbors Classifier

Finally, we trained a K nearest neighbors classifier (KNNC). The KNNC AUC was 0.873 (Figure 1D). The sensitivity, specificity, PPV, and NPV were 85.7%, 92.1%, 54.5%, and 98.3%, respectively (Table 3 and Figure 2). The F1-score and the accuracy were respectively 66.7% and 91.4%. The grid search cross validation took 38.5 s and it took 36 ms for the best model to fit.

### 3.6. Stacking Efficient Classifiers

#### 3.6.1. Classifier Metrics

In order to further improve model metrics, we chose to stack the previous three classifiers within an XGBoost classifier with the following parameters: ‘n_estimators’ = 10,000; ‘min_child_weight’ = 5; ‘min_sample_weight’ = 5; ‘max_depth’ = 7; and ‘learning_rate’ = 0.15. With those parameters, the sensitivity, specificity, PPV, and NPV were 71.4%, 100%, 100%, and 96.9% respectively (Table 3 and Figure 2). The F1-score and the accuracy were 83.3% and 97.1% respectively. The AUC of the ensemble model yielded an AUC of 0.899 (Figure 1E). The threshold value for the ‘predict_proba’ function that ensured the lowest number of misclassified patients (false negative + false positive) was 0.5.

#### 3.6.2. Variable Importance Using Shapley Values

The Shapley values for each patient and each variable is plotted in Figure 3.

The ten most important features for prediction in this algorithm were ferritin levels, lymph node enlargement, maximal body temperature, platelet count, serum glutamic-oxaloacetic transaminase (SGOT/AST) levels, underlying immunosuppression, hemoglobin level, BMI, C-reactive protein, and lymphocyte count. Here, we found that high ferritin levels, the presence of a lymph node enlargement, a higher maximal temperature, a low platelet count, high levels of SGOT/AST, and the presence of an underlying immunosuppression are associated with the prediction of HLH.

## 4. Discussion

This study is the first to date to propose a machine-learning approach to diagnose HLH. Hemophagocytic lymphohistiocytosis is a rare entity with a poor prognosis, which is even worse as the diagnosis delay increases [17]. Unfortunately, there is no “gold standard” for diagnosing HLH, and most of the time the diagnosis is based on the clinician’s decision, which is based on clinicobiological considerations and mostly confirmed with the evolution of the disease and/or with the treatment efficacy and/or available scoring solutions. Therefore, the main goal of early diagnosis in HLH is to avoid its complications, admission to an intensive care unit (ICU), and death. The two main available scores for HLH diagnosis are currently the HLH-2004 criteria and the Hscore [6,15]. External validation studies revealed that those scores had high sensitivity and specificity, which could vary depending on the scored population [7,17]. HLH-2004 criteria and Hscore perform better in critically ill patients. Hscore performs better in non-IMID-related HLH patients since the original study included only 3% of patients with IMID-related HLH and thus, this linear model is probably unsuitable for populations with a high proportion of IMID-related HLH [7,16]. These performances are to be modulated depending on the cutoff score that is chosen to define HLH when using Hscore. However, these two scores suffer from several limitations.

First, the HLH-2004 diagnostic criteria were based on an expert consensus on significant changes that are usually found in HLH in children. These revised criteria were adapted from the 1991 criteria by Henter et al. that were meant to be used as a classification tool for the HLH-94 prospective trial [29,30]. Thus, one should not consider the criteria as a diagnostic tool but rather a classification tool for research purposes. The criteria were arbitrarily equally weighted, and so were the number of criteria to be fulfilled to establish the diagnosis of HLH. However, the diagnostic performances of the HLH-2004 diagnostic criteria are probably insufficient in early-stage HLH where important markers such as bone marrow hemophagocytosis and cytopenias are usually lacking.

The Hscore resulted from a logistic regression model [15]. In this score, all quantitative variables were discretized. This choice may lead to information loss, especially for extreme values (i.e., in the case of extreme hyperferritinemia), but was essential for the deployment of the tool in order to be as simple as possible without significant performance loss. Although in external validation studies, the cutoff that was proposed by the authors failed to be sufficiently specific, yielding to high sensitivity, the logistic regression design allows to adapt the score cutoff to fit a given population [7,16,17]. Finally, the parameters that were selected to build this tool were chosen from a web-based Delphi study [31]. Another limitation could be the interpretation that a clinician would make of such a score. Indeed, the Hscore also provides a tool to predict the probability of having HLH in any patient. In fact, the given probability corresponds to the basal probability of a patient of the original Hscore dataset who fulfills none of the Hscore criteria which is:$$\frac{e^{-\left(intercept\right)}}{1+e^{-\left(intercept\right)}}$$

And this probability increases according to β coefficients and according to variables that are present in the tested patient:$$\frac{e^{-\left(intercept+\beta 1*\mathrm{var}1+\;\beta 2*\mathrm{var}2+\cdots \right)}}{1+e^{-\left(intercept+\beta 1*\mathrm{var}1+\;\beta 2*\mathrm{var}2+\cdots \right)}}$$

This probability represents the probability of having the H score compared with a patient without any symptoms in the Hscore original cohort, and should not be taken as an absolute probability of having HLH. Hence, clinicians should keep in mind that predictions or diagnostic tools have to be used on the precise population that they were built on (i.e., patients with at least HLH suspicion). In the initial study, there were more than 50% of patients with hematological malignancies. Although this is one of the limitations of the Hscore, any score or machine-learning model (including the present one) would have the same problem, making generalization a major pitfall for a model deployment for medical diagnosis especially without big data.

In the past decade, the democratization of machine learning through more accessible and popular programming languages such as Python or R allowed the spreading of various solutions for medical purposes. In the case of HLH, the main goal is to establish an early diagnosis in order to treat patients as soon as possible to avoid major complications (e.g., organ failure, ICU admission, death). Here, we propose a proof-of-concept approach for HLH diagnosis through the scope of machine learning. Our model reached high diagnostic performances in the test set and has the advantage to be possibly trained again to adapt to new data. Another advantage of this ML approach is the use of potentially unsuspected factors as important features for diagnosis, such as BMI. Moreover, in tree-based models, features are not analyzed one by one but in a succession of nodes that allows the combination of features to be considered for predictions. Finally, our model seems to outperform the Hscore in the test dataset in terms of misclassification rate: three patients being wrongly classified as HLH vs. zero in our model, and one patient not being diagnosed with HLH vs. two in our model. This increased performance compared with the Hscore is probably due to the difference in the patient selection process. Moreover, our model did not need any information on bone marrow, spleen, or liver hemophagocytosis for making predictions, contrarily to other available scores. Still, this should not be an argument for replacing Hscore with our model as it is only a proof-of-concept approach for HLH diagnosis at this stage. Our model provides clues for further considerations on diagnostic tools and metrics to be used to assess diagnostic performance of scores in HLH patients.

A limitation of our study is its retrospective nature, and thus all biases brought with it. In our case, missing values had to be imputed, which introduced a bias in the sense that using a specific pattern to replace missing values may influence the model predictions. To mitigate this issue, we did not impute and removed variables with more than 20% of missing values. Another limiting point was the inhomogeneous time at data collection. Indeed, every biological dosage was performed at the time of GF dosage and thus, in our data collection, the time was not taken into account. Such considerations in further studies may help to distinguish early markers for HLH detection.

Moreover, the retrospective nature of the study led us to select our patients through glycosylated ferritin dosage in patients from our center. This criterion makes our population a selected one since glycosylated ferritin is not measured in every patient with HLH suspicion. This may lead to three major issues. First, we expose ourselves to the risk of having a ratio of HLH (here 10%) which is not representative of the national incidence. This imbalanced ratio, combined with the fact of having a rather small number of observations from the machine-learning perspective, led us to use a SMOTE method to achieve better predictive performances. Without it, the trained models were constantly overfitting due to the overrepresentation of non-HLH patients. Indeed, the latter models were having a satisfying specificity and negative predictive value but poor sensitivity and positive predictive value. The use of SMOTE would differ on a larger cohort or with a different HLH ratio in the training set. Secondly, there is a potential bias toward more severe patients, thus making it perform worse on patients with HLH and mild symptoms or early disease. Finally, by training and testing our model only on a specific population (i.e., selected through biological testing), our model might badly generalize on ‘real-life’ populations.

Another limitation of this study is that HLH diagnosis was based on clinicians’ opinions. Yet no other approach would be entirely satisfactory in a condition for which, to date, there is no gold standard or consensus definition for sHLH in adults. This is why the PRINTO MAS score, the MS score, and the Hscore also used expert judgment as a gold standard [12,13,14,15]. Indeed, defining HLH cases by the existing classification criteria would exclude all conflicting cases from the analysis and prevent the evaluation of those same previous sets of criteria. However, the real added value of such a diagnostic tool is essentially based on improving the disease outcome (i.e., organ failure, ICU admission, or death). A machine-learning model built on a target in which the definition is based on expert opinion would only help to make a diagnosis as an expert would. Thus, our tool is only intended to provide an expert’s perspective to non-expert clinicians, not to replace an expert’s opinion in HLH diagnosis. As is the case for all machine-learning approaches, the model will only be able to reproduce the behavior suggested to it by the training data set. Thus, in the context of complex diseases, the algorithm will at best only be able to reproduce the decision resulting from an evaluation by experts, which is an interesting point if we place ourselves in the context of a diagnostic aid for non-experts. A better comparison for conditions without gold standard would be improvements in length or quality of life, such as in early-stage cancer [32].

## 5. Conclusions

We provided a proof-of-concept model for HLH prediction based on a machine-learning approach. As a proof of concept, this model is not intended to be deployed for a specific population but provides evidence that machine learning is likely to be the most powerful approach to medical diagnosis in the years to come for HLH and probably for other complex diseases. Further studies will provide a more generalizable diagnostic tool for HLH that would take into account more robust criteria such as survival in the absence of a gold standard for HLH diagnosis.

## 6. Patents

This section is not mandatory but may be added if there are patents resulting from the work reported in this manuscript.

## Figures and Tables

**Figure 1 jcm-11-06219-f001:**
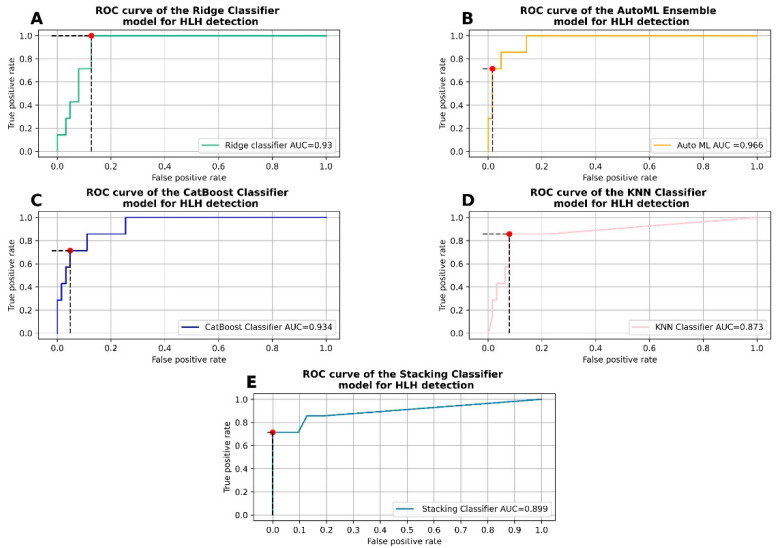
Receiver operating characteristic curves for the five different algorithms used for HLH detection. (**A**) AutoML ensemble model, (**B**) ridge classifier, (**C**) CatBoost classifier, (**D**) K nearest neighbors (KNN) classifier, and (**E**) stacking classifier with XGBoost as a meta-estimator. Red dots are used to mark the true positive rate and the false positive rate at the threshold used for predictions. Abbreviations: AUC: area under curve; HLH: hemophagocytic lymphohistiocytosis; and ROC: receiver operating characteristic.

**Figure 2 jcm-11-06219-f002:**
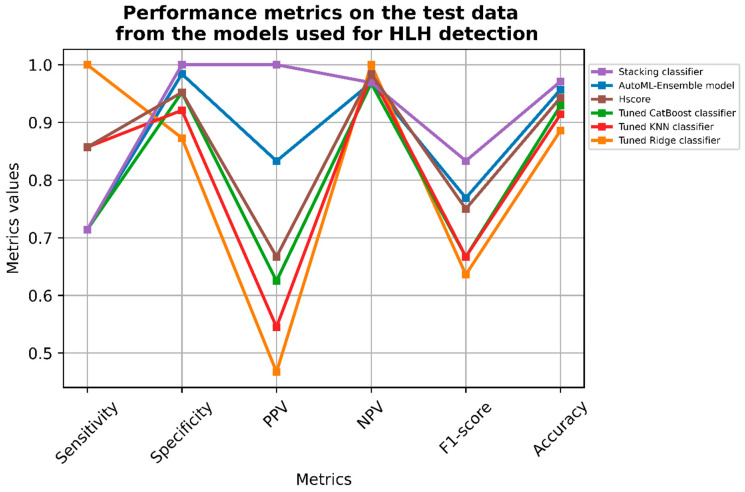
Line plot representing the main metrics values of each model used in the study. Abbreviations: KNN: K nearest neighbors; ML: machine learning; PPV: positive predictive value; and NPV: negative predictive value.

**Figure 3 jcm-11-06219-f003:**
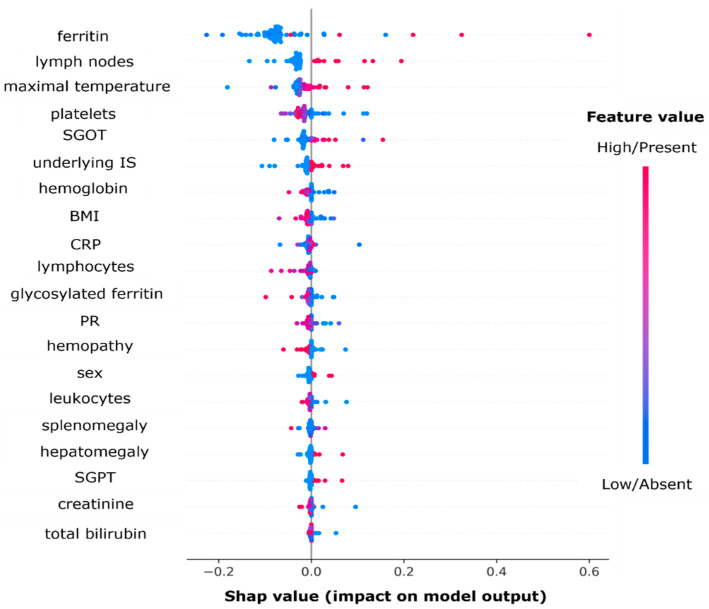
Beeswarm plot of Shap values for all patients and for each feature. The right side of the plot favors the output “HLH” and the left side favors the output “not HLH”. The gradient color corresponds to the feature value (dark pink if high or present for categorical features and blue if low or absent for categorical features). Abbreviations: BMI: body mass index; CRP: C reactive protein; PR: prothrombin rate; SGOT/AST: serum glutamic-oxaloacetic transaminase; and SGPT/ALT: serum glutamic-pyruvate transaminase.

**Table 1 jcm-11-06219-t001:** Summary description of the original dataset.

	Control	HLH	*p*. Overall
	*N =* 207	*N =* 24	
Clinical characteristics			
Sex (male)	122 (58.9%)	9 (37.5%)	0.074
Age at onset	53.0 [37.0; 65.5]	49.5 [32.8; 64.2]	0.562
Etiological subgroups			<0.001
Hematological malignancies	13 (6.28%)	11 (45.8%)	
Inflammatory diseases	90 (43.5%)	8 (33.3%)	
Infectious diseases	48 (23.2%)	3 (12.5%)	
Liver diseases	10 (4.83%)	0 (0.00%)	
Solid neoplasia	10 (4.83%)	1 (4.17%)	
Other	36 (17.4%)	1 (4.17%)	
Chronic liver disease	7 (3.38%)	2 (8.33%)	0.237
CKD	12 (5.80%)	1 (4.17%)	1.000
Underlying IS	69 (33.3%)	18 (75.0%)	<0.001
HIV	2 (0.97%)	2 (8.33%)	0.055
Solid neoplasia	18 (8.70%)	1 (4.17%)	0.701
Hematological malignancy	24 (11.6%)	9 (37.5%)	0.002
HLH-inducer treatment	56 (27.1%)	10 (41.7%)	0.207
Genetic predisposition	2 (0.97%)	1 (4.17%)	0.282
Physical exam features			
Max body temperature	37.7 [36.8; 38.9]	39.5 [39.0; 40.0]	<0.001
Hepatomegaly	17 (8.21%)	6 (25.0%)	0.020
Splenomegaly	15 (7.25%)	10 (41.7%)	<0.001
Lymph nodes	26 (12.6%)	15 (62.5%)	<0.001
Papular rash	36 (17.4%)	7 (29.2%)	0.170
Transient rash	6 (2.90%)	1 (4.17%)	0.541
Arthritis	37 (17.9%)	2 (8.33%)	0.386
Arthralgias	67 (32.4%)	5 (20.8%)	0.357
Odynophagia	25 (12.1%)	3 (12.5%)	1.000
BMI	24.0 [21.0; 27.0]	21.0 [20.0; 25.0]	0.166
Biology			
Ferritin	633 [276; 1618]	8749 [2402; 25,713]	<0.001
Glycosylated ferritin (%)	39.0 [26.0; 53.5]	23.5 [8.00; 34.0]	<0.001
TSR	0.16 [0.10; 0.31]	0.12 [0.09; 0.31]	0.628
CRP	71.0 [18.0; 155]	77.0 [29.2; 205]	0.203
PCT	0.50 [0.10;1.50]	4.00 [0.20; 10.2]	0.112
AST	30.0 [20.0; 54.0]	112 [67.0; 172]	<0.001
ALT	32.0 [18.2; 62.2]	79.0 [46.2; 160]	0.001
Total bilirubin	8.00 [6.00; 13.0]	15.5 [6.75; 33.8]	0.009
LDH	295 [220; 404]	590 [394; 946]	<0.001
CPK	43.0 [21.8; 87.5]	106 [31.0; 173]	0.138
Creatinine	68.0 [57.0; 81.0]	55.0 [45.5; 71.0]	0.017
Triglycerids	1.40 [1.00; 1.90]	2.85 [2.15; 3.97]	<0.001
PR	78.0 [67.0; 90.0]	74.5 [55.2; 90.2]	0.225
Fibrinogen	5.40 [3.27; 8.40]	2.75 [1.75; 4.75]	0.002
Hemoglobin	114 [93.0; 135]	90.0 [78.2; 100]	<0.001
Platelets	255 [184; 334]	62.0 [32.8; 138]	<0.001
Leukocytes	8.40 [5.60; 11.6]	2.30 [1.25; 4.93]	<0.001
Neutrophils	5.10 [3.25; 8.40]	1.55 [0.75; 3.20]	<0.001
Neutrophils (%)	0.67 [0.57; 0.77]	0.72 [0.54; 0.81]	0.730
Lymphocytes	1.60 [1.00; 2.10]	0.70 [0.32; 1.00]	<0.001
Evolution			
ICU	36 (17.4%)	10 (41.7%)	0.012
Death	17 (8.21%)	4 (16.7%)	0.248

Abbreviations: ALT/AST: alanine/aspartate aminotransferase; BMI: body mass index; CKD: chronic kidney disease; CPK: creatinine phosphokinase; CRP: C reactive protein; HIV: human immunodeficiency virus; HLH: hemophagocytic lymphohistiocytosis; ICU: intensive care unit; LDH: lactate dehydrogenase; PCT: procalcitonin; PR: prothrombin ratio; and TSR: transferrin saturation ratio. Medians and their interquartile ranges (brackets) were provided for quantitative variables.

**Table 2 jcm-11-06219-t002:** Area under the receiving operator characteristics curves (AUC) of different algorithms from the autoML pipeline on the training set.

Model Type	AUC	Train Time (s)
Ensemble	0.997475	0.47
CatBoost	0.994949	1.57
Linear	0.989899	1.88
Nearest neighbors	0.988636	1.66
Neural network	0.987374	1.22
LightGBM	0.984848	3.37
Xgboost	0.982323	3.71
Random forest	0.967172	3.73
Extra trees	0.959596	2.09
Decision tree	0.767677	2.5
Baseline	0.5	1.59

Abbreviations: AUC: area under the curve; GBM: gradient boosting machine.

**Table 3 jcm-11-06219-t003:** Performance metrics on the test data for the five different algorithms used for HLH detection. The highest metric value for each model is displayed in bold type.

	AutoML-Ensemble Model	Tuned Ridge Classifier	Tuned CatBoost Classifier	Tuned KNN Classifier	Stacking Classifier	Hscore
Sensitivity	0.714	**1.000**	0.714	0.857	0.714	0.857
Specificity	**0.984**	0.873	0.952	0.921	**1.000**	0.952
Positive predictive value	0.833	0.467	0.625	0.545	**1.000**	0.667
Negative predictive value	0.969	**1.000**	**0.968**	**0.983**	0.969	**0.984**
F1-score	0.769	0.636	0.667	0.667	0.8333	0.750
Accuracy	0.957	0.886	0.929	0.914	0.971	0.943
Threshold for probability	0.50	0.36	0.49	0.50	0.50	NA

Abbreviations: KNN: K nearest neighbors; ML: machine learning; PPV: positive predictive value; and NPV: negative predictive value.

## Data Availability

Data and Python script are available upon request to the corresponding author.

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
