# Peer review of "Diagnosing Hemophagocytic Lymphohistiocytosis with Machine Learning: A Proof of Concept"

_jcm, 2022, doi:10.3390/jcm11206219_

Round 1
Reviewer 1 Report
El Jammal et al. report a proof-of-concept study and provide a machine learning model for HLH detection in a retrospective cohort. They found good overall diagnostic performances with a sensitivity of 71.4% and a specificity of 100%. The topic is of high relevance, the study adds new aspects to the field, and is well-written. Some points need to be considered:
- - Introduction: „Several subsequent studies aimed at testing the validity of the Hscore on cohorts of patients with more varied triggers have all found poorer diagnostic performance. In particular, specificity was lower (specificity between 71-79%, depending on the study vs 86% in the original population)(7,15–17).“ Not all references show specificities of 71-79 %. However, these studies are of retrospective nature, which limits the value of validation of the Hscore in these cohorts and could also be mentioned. Further, the above mentioned sentences is in contrast to the discussions‘ sentence „External validation studies revealed that those scores had high sensitivity and specificity, which could vary depending on the scored population (7,17).“
- - What was the rationale for using 26 different variables?
- - Gender should be replaced by sex
- - The figures need higher resolution
- - The machine learning results are hardly to understand for readers who are not into machine learning interpretation
- - It would be intersting to know which variables predict for ICU admission and which for death within the HLH patients?
Author Response
Response to the reviewers :
Reviewer 1 :
“El Jammal et al. report a proof-of-concept study and provide a machine learning model for HLH detection in a retrospective cohort. They found good overall diagnostic performances with a sensitivity of 71.4% and a specificity of 100%. The topic is of high relevance, the study adds new aspects to the field, and is well-written. Some points need to be considered:
Introduction: “Several subsequent studies aimed at testing the validity of the Hscore on cohorts of patients with more varied triggers have all found poorer diagnostic performance. In particular, specificity was lower (specificity between 71-79%, depending on the study vs 86% in the original population)(7,15–17).” Not all references show specificities of 71-79 %. However, these studies are of retrospective nature, which limits the value of validation of the Hscore in these cohorts and could also be mentioned. Further, the above mentioned sentences is in contrast to the discussions‘ sentence „External validation studies revealed that those scores had high sensitivity and specificity, which could vary depending on the scored population (7,17).“””
We thank the reviewer for noticing such an issue. We adjusted the specificity values based on the references and added a statement about the limitations of the studies' retrospective nature.
Please find the modified sentences:
- Line 73 : “ In particular, specificity was variable (specificity between 71-93%[...])”
- Lines 76-77 : “Moreover, the external validation in these studies are limited by their retrospective nature.”
“What was the rationale for using 26 different variables?”
For the variable selection, we tested several hypotheses and found an optimal model accuracy with these 26 values. Removing additional values resulted in fewer accuracy.”.
Please find an explicative sentence line 166-8 : “Again due to performance loss, no variable among the 26 were removed from the final model.”
“Gender should be replaced by sex”
All the occurrences of “gender” were replaced by sex, accordingly.
“The figures need higher resolution”
Figure resolution was improved as requested.
“The machine learning results are hardly to understand for readers who are not into machine learning interpretation”
We thank the reviewer for raising this concern. To simplify for non-expert reader, we have added the definitions of F1-score and accuracy. Precision and recall were removed since they represent the same metric value as PPV and sensitivity, respectively. Those were typos.
Please find the modification line at 205-7 :
“To compare the different models on the test set, sensitivity, specificity, positive predictive value, negative predictive value, accuracy (true positive (TP)+ true negative (TN)/(positive (P)+ negative (N)), and F1-scores (2 x TP/(2 x TP + false positive + false negative)) were calculated.“
“It would be intersting to know which variables predict for ICU admission and which for death within the HLH patients?”
This is an interesting point raised by the Reviewer. Unfortunately, the initial purpose of our model was to determine if a patient has HLH or not (i.e., for diagnostic purpose). Outcome data were not recorded as this machine learning process was not built to predict ICU admission or death. Therefore, our model cannot do such predictions. We retain the idea for a further study, for which it would be necessary to build two additional models.
Reviewer 2 Report
Jammal and colleagues created a machine learning algorithm to try identify HLH patients. This is compared against 3 expert human opinions and the dataset used for machine learning consists of 26 variables of which they highlight 10 identified as most important, with ferritin being the most significant predictor.
The foremost point to note is that I am a biologist and know nothing of machine learning and as such half of this paper did not make sense to me.
That aside, this paper, like all other HLH prediction papers have the same shortcomings. As mentioned in the last discussion paragraph, the gold standard is clinician opinion, and would likely stay that way because there are too many variables (or missing variables) for the algorithm to take into account. Hence the results of this report shows what we already know: ferritin is an important marker. Even with this data hitting 100% specificity, the final clinical decision still lies in the hands of the clinician, regardless of which HLH prediction mechanism employed. The work performed can thus be concluded as a curious academic exercise.
Another issue I have with such studies is the timepoint in which the clinical parameters are collected. Are all samples collected at the same time of disease progression? How early in disease were the data collected? This is because the authors stated in line 328/388 that the goal was to have early diagnosis. Thus full blown clearly HLH samples should be excluded since those are ‘simpler’ cases to diagnose not needing a complex algorithm? It’s the very early timepoint or more challenging cases that we need help to diagnose, correct? HLH is a progressive disease and while the goal is to detect it as early as possible, no marker would work if it is too early in the disease course. And in Line 430 is it saying the system does not perform well for milder disease? Thus another suggestion I have for the authors is to collect longitudinal data as the rate of progression might be something useful.
Line 26, 75-76, 398-399. The point about consensus between machine predicted and expert opinion. How sure are we that the gold standard human decision is the truth? Could it be that the machine prediction is more accurate than our human mind and where machine scores differ from humans it could be a learning lesson for us rather than saying the machine is inaccurate?
The abstract should clearly state data was acquired on adult population.
Line 54: cytopenias may occur late but yet are included in the list on Line 160-161, even with a goal of early disease prediction? And still showed up on the top 10 hits. So does this mean cytopenias show up early enough to be an early HLH marker?
Line 61. Diagnosis of what specifically?
Table 1. data in different bracket types need to be explained in the legend.
Please cite for statement on lines 355-358.
Could BMI be interpreted as failure to thrive?
Could maximal temperature be interpreted as unremitting fevers?
Line 377: could you please point out the precise population this current tool can be used on?
Author Response
Reviewer 2 :
“Jammal and colleagues created a machine learning algorithm to try identify HLH patients. This is compared against 3 expert human opinions and the dataset used for machine learning consists of 26 variables of which they highlight 10 identified as most important, with ferritin being the most significant predictor. The foremost point to note is that I am a biologist and know nothing of machine learning and as such half of this paper did not make sense to me. That aside, this paper, like all other HLH prediction papers have the same shortcomings. As mentioned in the last discussion paragraph, the gold standard is clinician opinion, and would likely stay that way because there are too many variables (or missing variables) for the algorithm to take into account. Hence the results of this report shows what we already know: ferritin is an important marker. Even with this data hitting 100% specificity, the final clinical decision still lies in the hands of the clinician, regardless of which HLH prediction mechanism employed. The work performed can thus be concluded as a curious academic exercise.”
We thank the reviewer for his/her kind comments and careful reading of our paper although machine learning is not his/her field of predilection.
We have combined the two reviewer's comments as they were related and provide a comprehensive response below:
“Line 26, 75-76, 398-399. The point about consensus between machine predicted and expert opinion. How sure are we that the gold standard human decision is the truth? Could it be that the machine prediction is more accurate than our human mind and where machine scores differ from humans it could be a learning lesson for us rather than saying the machine is inaccurate?”
We fully agree that the final diagnosis remains the opinion/decision of the clinician, which is unfortunately always the case when there is no gold standard or specific sign/biomarker. Surely biologists and clinicians are much more comfortable in situations where these markers exist.
The main (and only) objective of our work was to provide proof of concept of an AI-based diagnostic aid. In the future, and after several optimization and validation steps (probably by including thousands of prospectively collected data), such a tool could be used to help non-expert physicians diagnose HLH (or include in clinical trials, or use as an “alert tool” incremented in patients' digital records).
Note that in our study, the diagnosis of HLH was made (or rejected) by clinicians who are experts in HLH. This methodology is the most commonly used in studies of diseases of complex diagnosis, such as HLH.
Thus, the tool does not provide a "true diagnosis" but rather best classifies each case in the light of the expertise of HLH experts.
The learning process of an ML algorithm aims to reproduce the structure of the training dataset on new data. Thus, if the label is provided by an expert, the algorithm can only aim to be as good as the expert (it is impossible, as the reviewer brilliantly points out, to know whether in some situation the algorithm was right and the expert was wrong).
Interestingly, unlike the usual diagnostic scores based on a restricted set of variables selected by a consensus of experts, ML-based tools can take into account some (or even many) variables that would not be identified as disease-specific by the expert (but that would intuitively influence his/her decision) because the variables are related to each other (e.g. BMI, age and sex influence the interpretation of ferritin: a high ferritin level does not have the same meaning in a young girl as in an obese old man).
Furthermore, the operation of these algorithms relies on the initial selection of the best performing parameters to guide their subsequent "path" (e.g. if ferritin is extremely high, the path followed will be the one based on the elements of the training set related to extreme ferritins etc.)
Finally, proof-of-concept studies are early steps towards the development of more powerful tools that are likely to assist clinicians in the near future (hopefully to the benefit of patients). Such tools may be especially useful in conditions such as HLH, in which delayed diagnosis has been shown to have a significant effect on morbimortality.
Indeed, the final decision for diagnosis rests with the clinician, as would any scoring system or biomarker combination for any syndrome or complex disease. Here, the purpose of this work is to provide a proof of concept for a diagnostic tool that would be meant to help the non-expert practitioner diagnose HLH. The “target” variable (i.e., HLH or not HLH) was collected according to the expert opinion as in other HLH diagnostic score papers. Here, the tool is not providing a true diagnosis but expert knowledge about HLH. The training process of a machine learning algorithm is meant to reproduce the training dataset structure on new data. Thus, if the label is provided by an expert, the algorithm will only be capable of being as good as the expert would be.
In order to ease the comprehension of this point, we added :
“Thus, our tool is only intended to provide an expert’s perspective to non-expert clinicians, not to replace an expert’s opinion in HLH diag-nosis. As is the case for all machine learning approaches, the model will only be able to reproduce the behavior suggested to it by the training data set. Thus, in the context of complex diseases, the algorithm will at best only be able to reproduce the decision resulting from an evaluation by experts, which is an interesting point if we place ourselves in the context of a diagnostic aid for non-experts..” Line 457-65:
Another issue I have with such studies is the timepoint in which the clinical parameters are collected. Are all samples collected at the same time of disease progression? How early in disease were the data collected? This is because the authors stated in line 328/388 that the goal was to have early diagnosis. Thus full blown clearly HLH samples should be excluded since those are ‘simpler’ cases to diagnose not needing a complex algorithm? It’s the very early timepoint or more challenging cases that we need help to diagnose, correct? HLH is a progressive disease and while the goal is to detect it as early as possible, no marker would work if it is too early in the disease course. And in Line 430 is it saying the system does not perform well for milder disease? Thus another suggestion I have for the authors is to collect longitudinal data as the rate of progression might be something useful.”
The clarifications you request are perfectly relevant. In fact, our inclusion criteria and the retrospective nature of the study did not allow us to collect samples at the same stage of disease progression.
Patients were included for the sole reason of having a glycosylated ferritin assay (marker of HLH), which could occur at any time during the course of the disease. This depends on the timing of the assay request by the clinician in charge.
The specific objective of this work was not to study only patients in the early phase of HLH. This is not what we announced in the method section. The comment about the need for early diagnosis in HLH is a general one. Developing a diagnostic model, whatever the stage of the disease, can help in the end for this general purpose. This is particularly true for clinicians who are not used to this disease, and for whom the diagnostic expertise partially reproduced through our machine learning model would help them to make faster diagnoses. This includes patients with a well-established HLH, in front of whom they could still have diagnostic doubt. It would not have been relevant, in this preliminary proof of concept study, to exclude samples from patients with a well-established disease. Indeed, these patients are useful to the A.I. for training in making diagnoses. Our work is only a first transitional step towards a more precise objective of detecting early forms. Your suggestion to include patients in the very early phase and to integrate dynamic parameters from longitudinal data is thus very relevant, and will have to be the subject of further research.
In order to ease the comprehension of this point, we added :
“Another limiting point was the inhomogeneous time at data collection. Indeed, every biological dosage was performed at the time of GF dosage and thus, in our data collection, the time was not taken into account. Such considerations in further studies may help to distinguish early markers for HLH detection.” (line 422-line 426)
“Line 26, 75-76, 398-399. The point about consensus between machine predicted and expert opinion. How sure are we that the gold standard human decision is the truth? Could it be that the machine prediction is more accurate than our human mind and where machine scores differ from humans it could be a learning lesson for us rather than saying the machine is inaccurate?”
This comment is in line with the first one, and again, this model is intended to provide expert opinion to a non-expert. Our machine learning model works only in supervised learning and cannot produce a different diagnostic understanding than the expert who classified the patients. It only copies his clinical judgment. This prediction tool is only capable of being as accurate as an expert would be. In such difficult diseases, the goal is to make expert opinion easily accessible to any clinician, so any model or scoring system should strive to correlate as closely as possible with expert opinion.
“The abstract should clearly state data was acquired on adult population.”
The abstract was modified accordingly:
“On a dataset of 207 adult patients with 26 variables […] ”
“Line 54: cytopenias may occur late but yet are included in the list on Line 160-161, even with a goal of early disease prediction? And still showed up on the top 10 hits. So does this mean cytopenias show up early enough to be an early HLH marker?”
In Figure 3, we provide the shapley values for our model. Here, the variables voluntarily remained non-discretized and were provided as blood cell counts (/mm3). This allows the model to interpret each input value with regard to all other variables in the same individual. This plot only refers to the importance of each variable for the model’s prediction. Nevertheless, the retrospective nature of the study did not allow us to provide biological data at a precise time point, but, like the “materials and methods” paragraph states, biological data were collected at the time of glycosylated ferritin dosage. This is also one limitation provided by the retrospective nature of the data collection.
Such an issue was taken into account as follows:
“Another limiting point was the inhomogeneous time at data collection. Indeed, every biological dosage was performed at the time of GF dosage and thus, in our data collection, the time was not taken into account. Such considerations in further studies may help to distinguish early markers for HLH detection.” (line 422-line 426)
Line 61. Diagnosis of what specifically?
We are referring to HLH diagnosis delay :
“The median time delay to HLH diagnosis is 10 days” line 61.
“Table 1. data in different bracket types need to be explained in the legend.”
Modifications were made in the legend of table 1.
“Please cite for statement on lines 355-358.”
The citations for external validation studies were added.
“Could BMI be interpreted as failure to thrive? Could maximal temperature be interpreted as unremitting fevers?”
Here, we could not decipher such considerations. BMI probably reflects a failure to thrive in some patients, but no conclusions can be provided on this specific point. The maximal temperature is the highest temperature measured in the two days following or preceding the ferritin determination.
“Line 377: could you please point out the precise population this current tool can be used on?”
Since this study is a proof of concept study, this tool is not intended to be deployed at this time. Such a consideration was precised on line 470-4:
“As a proof of concept, this model is not intended to be deployed for a specific population but this provides evidence that …”